# Seeking ambulance treatment for 'primary care' problems: a qualitative systematic review of patient, carer and professional perspectives

Matthew J Booker, Sarah Purdy, Alison R G Shaw

► Prepublication history and additional material are available. To view these files, please visit the journal online (http://dx.doi.org/10.1136/bmjopen-2017-016832).

Centre for Academic Primary Care, School of Social and Community Medicine, University of Bristol, Bristol, UK

**Correspondence to**
Dr Matthew J Booker;
Matthew.Booker@Bristol.ac.uk

## ABSTRACT

**Objectives** To understand the reasons behind, and experience of, seeking and receiving emergency ambulance treatment for a 'primary care sensitive' condition.

**Design** A comprehensive, qualitative systematic review. Medline, Embase, PsychInfo, Cumulative Index of Nursing and Allied Health, Health Management Information Systems, Healthcare Management Information Consortium, OpenSigle, EThOS and Digital Archive of Research Theses databases were systematically searched for studies exploring patient, carer or healthcare professional interactions with ambulance services for 'primary care sensitive' problems. Studies using wholly qualitative approaches or mixed-methods studies with substantial use of qualitative techniques in both the methods and analysis sections were included. An analytical thematic synthesis was undertaken, using a line-by-line qualitative coding method and a hierarchical inductive approach.

**Results** Of 1458 initial results, 33 studies met the first level (relevance) inclusion criteria, and six studies met the second level (methodology and quality) criteria. The analysis suggests that patients define situations worthy of 'emergency' ambulance use according to complex socioemotional factors, as well as experienced physical symptoms. There can be a mismatch between how patients and professionals define 'emergency' situations. Deciding to call an ambulance is a process shaped by practical considerations and a strong emotional component, which can be influenced by the views of caregivers. Sometimes the value of a contact with the ambulance service is principally in managing this emotional component. Patients often wish to hand over responsibility for decisions when experiencing a perceived emergency. Feeling empowered to take control of a situation is a highly valued aspect of ambulance care.

**Conclusions** When responding to a request for 'emergency' help for a low-acuity condition, urgent-care services need to be sensitive to how the patient's emotional and practical perception of the situation may have shaped their decision-making and the influence that carers may have had on the process. There may be novel ways to deliver some of the valued aspects of urgent care, more geared to the resource-limited environment.

## Strengths and limitations of this study

► This evidence synthesis expressly focuses on the experiences of providing and receiving ambulance care for 'primary care sensitive' clinical problems, which are forming an increasing part of prehospital care workload.

► The methods of qualitative systematic review build on previous literature mapping work to offer a more nuanced appreciation of the complexities experienced by patients, carers and professionals in seeking ambulance treatment.

► The majority of qualitative evidence is drawn from ambulance systems in more economically developed countries with an established primary care model and in adult populations.

## INTRODUCTION

Despite origins as services for those with acute medical emergencies and injuries,[1] the majority of contacts with ambulance organisations are no longer for serious immediately life-threatening conditions.[2] Calls to the emergency ambulance service in the UK have been rising over recent years at 7% per annum,[3] and are increasingly for conditions that could potentially be managed by a primary care provider.[4] The proportion of ambulance calls for mental health conditions and social situations are also rising for reasons that are poorly understood.[4]

A recent systematic mapping review by the authors identified a number of factors that may be associated with ambulance use for problems that could be managed in primary care, including markers of deprivation, minority status and certain social circumstances.[5] Much of the previous literature exploring help seeking for such problems in the context of ambulance use has focused on the so-termed 'inappropriate' use of the service, usually defined from the healthcare professional perspective. However, defining

use in this way neglects to appreciate the complex processes that are likely to underlie help-seeking behaviours and attitudes in these user groups. Indeed, healthcare professionals themselves lack consensus as to what constitutes 'inappropriate' use of emergency healthcare.[6] There is increasing acceptance that labelling such ambulance services use as inappropriate fails to recognise the context of the request for assistance and is unhelpful for developing practical solutions. Additionally, making the decision to call an ambulance is not always the 'easy' option for many service users, as previous qualitative research by the authors[7] suggests that many emergency calls are preceded by a substantial degree of internal conflict on the part of the caller about the best course of action.

An established body of sociological research has sought to unpick what underpins illness behaviour and help seeking in general. The more established models of illness and help seeking acknowledge a complex interplay between biological predisposition to illness, experienced symptomatology, learnt behaviour patterns, attributional predispositions, situational influences and the organisational incentives and secondary benefits of the healthcare system itself.[8] Recent work examines how help-seeking and decision-making models may apply to some urgent care settings, such a GP out-of-hours[9] and emergency departments,[10] but there has been a lack of attention to how patients and carers conceptualise the need for ambulance care specifically. There is very limited understanding of how help-seeking models apply to situations where the request for ambulance care is for a problem that is potentially manageable in a primary care setting.

This paper, therefore, seeks to provide a more detailed and nuanced account of the experience of seeking and receiving ambulance treatment for 'primary care sensitive' conditions, by synthesising relevant data from qualitative studies of patients, carers and professionals.

## METHODS

An electronic systematic literature search was undertaken on the following databases: Medline, Embase, PsychInfo, Cumulative Index of Nursing and Allied Health, Health Management Information Consortium and Health Management Information Service. Google Scholar and Web of Science searches were undertaken to identify reports not captured by the above. Additional searches of OpenSigle, EThOS and Digital Archive of Research Theses databases were undertaken. The comprehensive systematic strategy was complemented by hand searches of key journals. This literature search formed the basis of a systematic literature mapping exercise,[5] the protocol of which has been published on the Prospective Register of Systematic Review Protocols register (reference CRD42014009108). Search terms were developed iteratively by consensus discussion among the research team and a medical subject librarian. An example search strategy is included as (see online supplementary appendix 1).

Initial searches were run for papers published between January 1980 and June 2014, with further update searches covering the period June 2014 to January 2017.

### Inclusion and exclusion criteria

The inclusion criteria for the systematic mapping review[5] were articles published in the English language, reporting the findings of research exploring patient, carer, healthcare professional or health service management perspectives and experiences of interaction with ambulance services for 'primary care sensitive' problems. The definition of a 'primary care sensitive' condition could be from any perspective, assigned prospectively or retrospectively. A 'primary care sensitive' contact could include explicit reference to terms related to primary care or family medicine or could be defined by reference to a comprehensive list of indicator conditions developed in conjunction with a medical subject librarian. Studies that reported on any stage of the interaction (from telephone call to treatment or transport) were of interest. Studies reporting on routine primary care only were excluded.

For this qualitative systematic review, further inclusion criteria were applied to identify papers reporting qualitative research. Papers were included if the data collection and analysis were wholly qualitative. Mixed-method papers were included if they reported identifiable components that had used qualitative data collection and analysis methods. Papers were screened by two researchers (MB, AS) independently, with all excluded papers further reviewed by all researchers to ensure agreement.

### Extracting, coding and analysing the data

A thematic synthesis was undertaken, following the approach described by Thomas and Harden.[11] All verbatim text in the 'results' and 'discussion' sections (or equivalent) of papers was treated as data for the purposes of this analysis. Data were loaded electronically into NVivo V.11 software and a process of line-by-line open coding undertaken. Results, which often included quotations or descriptions of themes, were coded alongside authors' interpretive statements in the discussion section. The synthesis took the form of a three-level inductive thematic analysis: open line-by-line coding of the primary studies, organisation into groups of descriptive themes and finally the development of overarching analytical themes.[12] The Enhancing Transparency in Reporting the Synthesis of Qualitative Research statement was followed throughout the data collection, appraisal and synthesis.[13]

There are a range of methodological approaches to handling and analysing data as part of a qualitative synthesis,[13] including metatheoretical and metaethnographic approaches that draw on grounded theory and line-of-argument principles to synthesise 'key concepts' (eg, Campbell et al[14]) and critical interpretive methods resulting in synthetic constructs. (eg, Dixon-Woods et al[15]) The line-by-line thematic synthesis approach[11] was used in this analysis. With philosophical foundations in critical realism, this approach permits a holistic understanding of the described phenomena through a more flexible way

**Table 1** Example data extracts demonstrating the hierarchical coding process

| Original text | Free codes | Descriptive themes | Analytical themes |
|---|---|---|---|
| "That's the worst of staying out here in the wilds out here because they speak about this 9 min and stuff o' this kind, but that is impossible staying oot here, like…" | 3.45 Difficult/impossible for ambulances to arrive quickly/ meet response time targets in geographically remote areas<br>3.46 Acknowledgement that living in rural area necessarily has (disadvantage) of time delay in getting emergency care | Rurality and remoteness: there are consequences of living in a rural or remote area in terms of how quickly ambulance care can be accessed. Patients recognise and even accept this constraint and adapt their expectations and the way they decide to physically access care and the way in which they get to the location of treatment. | Practical domain: whatever the perceived health need may be, and regardless of who may be involved assessing it, there are physical practicalities that sometimes override all other aspects of the decision-making process and take precedence. These can be related to geography and space, access to modes of transport, physical limitations of an individual's capability to care for themselves or the perceived need for immediate, expert care that can only be provided by ambulance staff. |
| "If it was very severe, I would get my husband… to drive me to the hospital." | 3.72 Very severe illness requires hospital treatment<br>3.73 Assistance from relative required to access appropriate care<br>3.74 Decision to drive/make own way to hospital instead of call and wait for ambulance | | |
| "I could have gone in my car… I would have made a lot of work for my relatives, which I don't think is right." | 2.92 Health condition such that could have gone by car but active decision not to due to convenience<br>2.93 Assistance from relative required to access care<br>2.94 Reluctance to inconvenience relatives as a result of own illness | Transport: role of seeing the ambulance service as a transport when other options are discounted as unsuitable or inconvenient, even when they may be entirely suitable as the clinical condition is relatively minor | Process domain: being influenced by others—perception of whether one would be a burden on one's relatives shapes the decision-making process about how to access urgent treatment (subtle or 'invisible' influence of relatives). |

that 'free' first-level codes can be applied to various types of source data in the included studies. Table 1 provides example extracts from the analysis to illustrate this process of coding 'up' from free codes, to descriptive then analytical themes. MB led on this process with regular research meetings between all authors to verify coding and agree themes by consensus.

### Assessment of quality

The assessment of quality in qualitative syntheses is much debated and there is little consensus about whether to do quality appraisal and how to do it.[11] Some qualitative methodologists argue that quality should not or cannot be meaningfully measured at all,[16 17] particularly if the synthesis aims to include all relevant existing data, however collected or reported. The authors of this synthesis take the view that while relevance of a paper to the review may outweigh reporting quality, it is helpful to attempt to assess quality in a consistent and transparent manner, to allow the reader to make judgements about how particular papers have contributed to a synthesis. In this synthesis, while studies were not excluded on the basis of a quality assessment alone, recognition of the quality limitations of included papers enabled a sensitivity analysis to be performed as described below.

A number of tools have been developed to assist in the systematic appraisal of quality in qualitative research.[18] Many of these are limited to informing the inclusion/ exclusion process, and some have been criticised for blending different aspects of quality assessment under too few headings. The Critical Appraisal Skills Programme (CASP) checklist[19] is often utilised for quality assessment in qualitative syntheses, prompting assessment of a paper against a number of items related to the purpose, design, conduct and reporting of qualitative research. A modified CASP checklist was used in this synthesis to assess included papers under a number of headings: overall appropriateness of the qualitative methodology, credibility, transferability, dependability and confirmability, including detail of the reporting. While there is considerable debate about assigning numerical 'scores' to such quality appraisals, the synthesis team adopted the four-tiered quality summary score described by Downe[20] to report the output of this quality assessment in a transparent manner. The final quality score, described in table 2, was assigned

**Table 2** Quality summary scores for qualitative studies (adapted from Downe[20])

| Score | Description |
|---|---|
| 1 | No or few flaws: credibility, transferability, dependability and confirmability is high. |
| 2 | Some flaws: unlikely to affect the credibility, transferability, dependability or confirmability. |
| 3 | Some flaws, which may affect the credibility, transferability, dependability and/or confirmability |
| 4 | Significant flaws, which are very likely to affect the credibility, transferability, dependability and/or confirmability |

by consensus among the research team following the appraisal process described above.

To enable judgement about how individual papers of varying reporting quality had contributed to the synthesis, a form of sensitivity analysis was undertaken as described by Thomas and Harden,[11] whereby the themes arising from each of the included papers were suppressed from the analysis. This was achieved by removing all coded data from the NVivo V.11 dataset, paper by paper and comparing the model against the full analysis. There was no substantial impact on the top-level conceptual model as a result of this, indicating that the final model was not disproportionately shaped by studies of any particular quality.

### The quality–relevance (Q-R) plot

Assessing relevance of qualitative studies in a synthesis is also an area of some debate and often includes assessment of some parameters that could arguably overlap with measurements of quality.[16] For the purposes of this qualitative synthesis, the concept of relevance is defined as applicability of the study's findings to the authors' health system and infrastructure, namely a (broadly) nationally homogeneous ambulance service operating within a health infrastructure with an established primary-care model. As such, relevance was assessed—again by consensus—into one of three categories, as summarised

| Table 3 | Categorisation of study relevance |
|---------|-----------------------------------|
| A | Concerning UK ambulance services and/or UK 'primary care' model |
| B | Concerning Westernised (but non-UK) ambulance service and/or non-UK 'primary care'-based model of healthcare |
| C | Concerning non-Westernised ambulance service and/or non-primary care-based model of healthcare. |

in table 3. To visualise the spread of included studies by both quality and relevance, the authors developed the concept of the quality–relevance (Q-R) plot (figure 1).

## RESULTS
### Search results

The literature search identified 1458 references (n=1424 during the initial search and n=34 in the update search). After duplicate suppression and removal of irrelevant and incorrectly cited documents, first-level screening against the inclusion/exclusion criteria resulted in 33 studies remaining (n=31 from the initial search and n=2 from the update search). Second-level screening for qualitative methods content resulted in six papers (n=5 from the initial and n=1 from the update searches) meeting

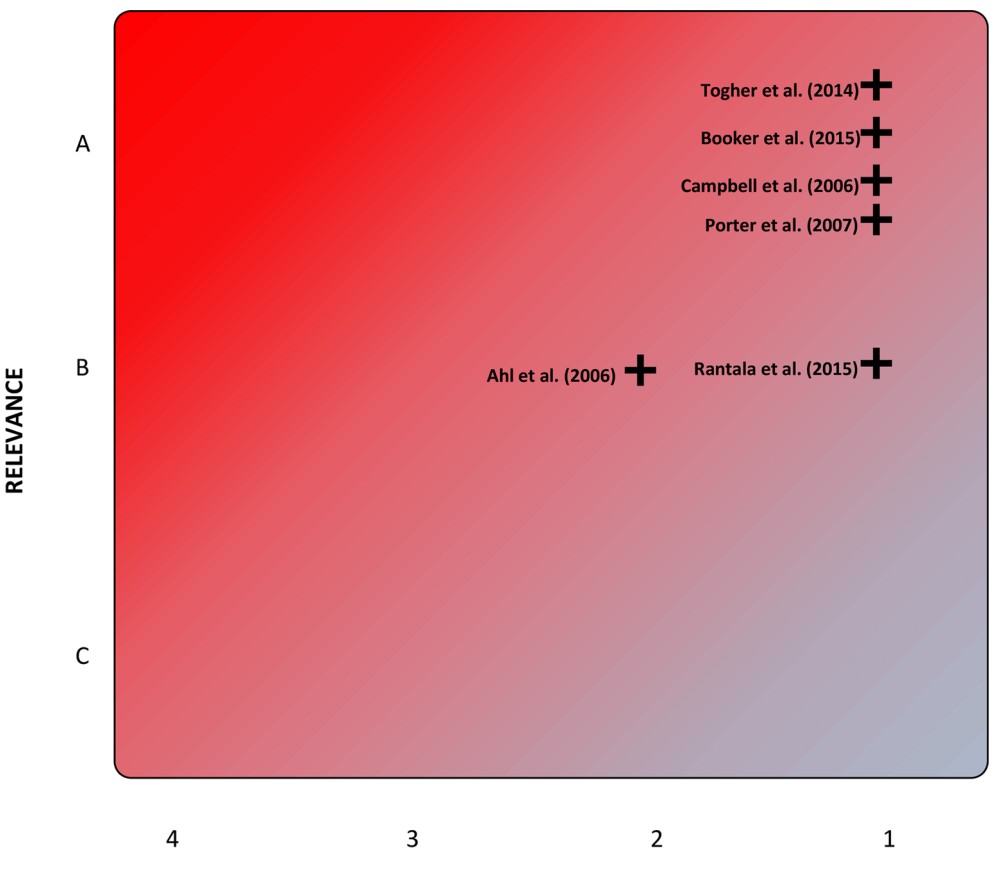

**Figure 1** 'Quality–relevance plot' of included studies.

**Table 4** Characteristics of the studies included in the thematic synthesis

| Paper | Year | Setting | Sample | Study methodology |
|---|---|---|---|---|
| Booker et al[7] | 2013 | UK ambulance service | 16 adult participants (patients and carers) | Qualitative semistructured interview study; thematic analysis |
| Ahl et al[28] | 2006 | Swedish ambulance service | 20 adult participants (patients) | Qualitative interview study, content analysis |
| Campbell et al[29] | 2006 | Scottish primary care | 78 adult participants (patients) | Qualitative semistructured interviews and focus groups, inductive thematic analysis |
| Rantala et al[30] | 2015 | Swedish ambulance service | 12 adult participants (patients) | Qualitative open-ended interviews, inductive phenomenological hermeneutic analysis |
| Porter et al[31] | 2007 | UK ambulance service | 25 adult participants (paramedics) | Qualitative focus group study; thematic analysis |
| Togher et al[32] | 2014 | UK ambulance service | 30 (22 patients, eight spouses) | Qualitative interview study, thematic analysis and mapping |

full inclusion criteria for the qualitative synthesis. The characteristics of these studies are summarised in table 4. Figure 1 summarises the authors' assessment of quality and relevance as described above.

### From open codes to descriptive themes

Following the process of free coding, a total of 23 descriptive themes were developed iteratively by repeated cycles of reductive grouping of codes until no additional discrete categories were required to fully describe the free-code dataset. As both verbatim primary data transcriptions and authors' interpretations were included as 'data' in this exercise, these descriptive themes already have a degree of interpretation included in them. These descriptive themes were then organised into nine, related descriptive theme groups. The relationship is shown in table 5.

### Development of analytical themes

By analysing patterns in the free codes and descriptive themes within and across the nine thematic groups, a number of cross relationships between groups were identified. Through a process of comparing the theme groups and their constituent descriptive themes, three overarching analytical themes were identified. These analytical themes conceptualise 'domains' of experience that are shared by all involved in the event of ambulance treatment for primary care sensitive conditions: practical domain, emotional domain and process domain.

The practical domain contains descriptive themes relating to the practicalities of needing and receiving treatment, including themes associated with access issues and transport. The emotional domain brings together descriptive themes concerning the emotions involved in needing, calling for, receiving and providing ambulance treatment. The process domain is a conceptually distinct and central domain, referring to the processes by which patients, carers and professionals try to integrate the practical needs and limitations of the situation with the emotional challenges associated with illness and help seeking. Figure 2 summarises how the three analytical themes are related.

### The practical domain

This analytical theme unites descriptive themes relating to the practicalities of asking for or providing ambulance care. While the practical aspects (including logistic difficulties and convenience) may not always be the first consideration, many of the themes include some aspect of patients and carers 'weighing up' how practical the use of the ambulance service (or alternatives) are for their perceived needs.

In many instances, patients appear to know, broadly, what is wrong with them and what treatment is required, but do not appreciate that the problem is out-of-scope of the ambulance service. Sometimes patients appear to genuinely expect the ambulance service to treat minor ailments and injuries in preference to GP services—a belief that appears rooted in misunderstanding of how urgent care services are structured. In these circumstances, the call to the ambulance service is made in 'good faith' that this is the right way to access treatment. Sometimes, however, patients acknowledge that the condition is not life threatening, but feel that there are genuine practical issues that would prevent appropriate treatment elsewhere, in what they see as a logical and reasoned attempt to arrive at the most sensible outcome.

> The doctor would find it very difficult to do anything useful, I think, just with me describing the symptoms of a water infection over the telephone. (Booker et al,[5] participant quotation)

> The doctor on the phone said [it may be] early appendicitis. You are bound to need blood tests and scans and stuff… so I just called the ambulance to get straight up to hospital. (Booker et al,[5] participant quotation)

The logic behind these decisions is not fundamentally unsound, but there is a lack of understanding of the implications of these access choices on the wider system. Practical issues appear to be viewed in differing ways, with often minor logistic barriers being

| Table 5 | Relationship between descriptive themes and thematic groups |
|---|---|
| Receiving treatment for a complaint that the caller knows is minor or non-serious | Needing and receiving treatment urgently |
| Receiving immediate, life-saving treatment for a condition the caller suspects is serious or life threatening | |
| Skill set and capabilities of the ambulance staff providing treatment | |
| Using the ambulance as a means of quick transport to hospital (when there is a medical need) | The transport role of the ambulance service |
| Using the ambulance as a 'convenient' mechanism of transport to hospital (with or without medical need) | |
| The impact of rurality and geographic remoteness on the care available when needed | Rurality and remoteness and the impact on receiving ambulance care |
| The impact or rurality and geographic remoteness on personal expectations and choices about accessing care | |
| Patient uncertainty about the seriousness of their health conditions and whether there is urgent medical need | Uncertainty about own health and the choice for ambulance care |
| Patient uncertainty about how to access the most appropriate healthcare service | |
| Patient uncertainty about the impact of their actions on others | |
| Uncertainty about making a decision on behalf of someone else (relative or friend) | |
| Need for personal reassurance that there is no underlying serious health problems | Reassurance about the absence of serious health problems |
| Need for reassurance that there is a legitimate need to access ambulance care | |
| Providing information to medical professionals to allow them to make appropriate decisions | Taking or handing over control of the situation |
| Taking decisive action to manage an intolerable circumstance | |
| Compassion provided by ambulance staff (in comparison to other healthcare groups) | Experience of compassion during contact with the ambulance service |
| Compassion of friends and relatives in times of illness when calling an ambulance | |
| Influence of others (friends and relatives) when asked for their opinion | Being influenced by the opinions and experience of others about ambulance care |
| Influence of others (friends and relatives) through their perceived opinions, even though the person may not present or directly consulted | |
| The need to take responsibility for one's own healthcare | Taking responsibility and being empowered to manage one's urgent health needs |
| The need to take responsibility for the healthcare of another (friend or relative) | |
| Empowerment to manage own healthcare | |
| Empowerment to access ambulance care appropriately | |

used as a justification for using ambulance services. The theme of the ambulance service fulfilling a transport role is one such example, with the idea of getting to hospital more quickly or more easily being central. There are examples of when patients articulate that ambulance transport is more convenient, if not strictly a necessity.

> I could have gone in my car… I would have made a lot of work for my relatives, which I don't think is right. (Ahl et al,[28] participant quotation)

Thus, while not always overt in the primary data, there were connections between data within the descriptive theme about 'convenience' of ambulance service use and data within the descriptive theme relating to perceived impact on others around the patient. Thus, 'convenience' extends to include patients not wishing to be a burden on their carers and relatives.

Convenience can also be viewed from the perspective of physical and geographic isolation. Being geographically remote was something that some patients acknowledged as impacting on what kind of service they might expect from the ambulance.

> That's the worst of staying out here in the wilds out here because they speak about this nine min and stuff o' this kind, but that is impossible staying oot here, like. (Campbell et al,[14] participant quotation)

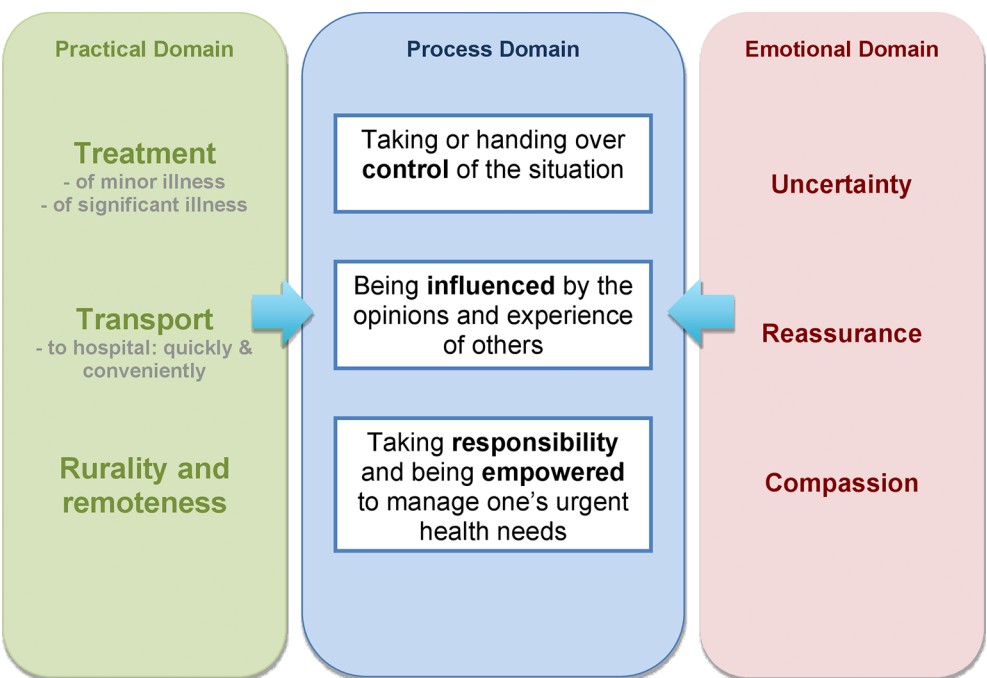

**Figure 2** Relationships between analytical themes.

This differing geography resulted in those living in urban areas taking a notably different course of action to those living in rural areas, seemingly as a result of the overt realisation of the impact of the 'remoteness' on the likelihood of getting a timely ambulance response.

> If it was very severe, I would get my husband… to drive me to the hospital. (Ahl et al,[28] participant quotation)

> Patients in rural areas consult primary healthcare less frequently than urban patients… and are more likely to call their own general practitioner in emergencies. (Ahl et al,[28] author interpretation)

Themes in the practical domain also allude to an undertone of frustration that patients can experience if they do not initially receive the care that they expect. Being unable to 'get through' to the right person first time seems to link with the negative emotional responses associated with inability to get reassurance, worsening uncertainty and anxiety and a feeling that the 'system doesn't care'. Free codes relating to practical barriers and access issues frequently coexisted with codes describing negative emotions:

> The walk in centre just sends you up to hospital if they can't do anything. So you end up, like, waiting 3 hours twice! (Booker et al,[5] participant quotation)

> The district nurse said she couldn't help. The doctor's surgery couldn't help. The hospital clinic couldn't help. After that many calls, you just snap? (Booker et al,[5] participant quotation)

Within the emotional domain, there were also descriptive themes and codes relating to the potential treatment that the ambulance service could provide, if necessary. Strong links existed between patients' knowledge of practical treatment that crews could provide, and the emotional themes of reassurance and uncertainty. This suggests that it is as much the potential and perceived capacity of the ambulance service to deliver life-saving treatment that is valued, and alludes to the idea of risk avoidance or risk minimisation when selecting which urgent care service to access.

### The emotional domain

This domain encompassed descriptive themes relating to how people responded emotionally to the experience of needing or receiving ambulance treatment. Descriptive themes in this domain were organised into three groups: the issue of uncertainty about their condition or health, the need for (and impact of) reassurance and the effect of compassion shown by others during the ambulance service contact. Analysis of emotion-related themes across the studies supports the ideas that the emotional response was not always congruent with actual medical need.

The descriptive theme of uncertainty was often associated with codes referencing anxiety but appears to more broadly reflect the idea of dealing with the unknown.

> I just needed someone there quickly… to tell me whether it was something serious? (Booker et al,[5] participant quotation)

> A sense of insecurity and vulnerability emerged due to now knowing what was wrong. (Rantala et al,[30] author interpretation)

Within the descriptive theme of uncertainty there existed a continuum of emotions, ranging from codes associated with minor indecision through to despair. There was no clear or consistent link with the actual type or severity of clinical condition experienced by these patients,

suggesting that extreme emotional responses were apparent in even relatively minor medical problems. Importantly, uncertainty appears to be experienced by ambulance crews as well, particularly around how best to manage non-urgent presentations.

> Even where the crew did not think that the patient should go to hospital and the patient did not want to go, crew members were still bothered by the fear of a possible comeback if it turned out to be the wrong decision. (Porter *et al*,[31] author interpretation)

This is particularly noteworthy, as the descriptive theme of reassurance links equally with themes that rest principally in the practical domain and themes that appear purely emotion driven. This suggests the potential powerful effect of reassurance—perhaps almost therapeutic—on helping patients gain practical and emotional control of the situation they are experiencing.

> I felt so ill that I thought I was about to become unconscious, but knowing that someone is on the way meant I was not worried, I knew they would find me. (Ahl *et al*,[28] participant quotation)

> The stress and pain began to ease when I knew that the ambulance was on its way. (Ahl *et al*,[28] participant quotation)

Also closely linked with reassurance and uncertainty is the descriptive theme of compassion. Codes within the compassion theme sometimes occurred alongside codes within the practical 'ambulance as transport' theme, but in a manner that suggests at least some patients do see these two roles as distinct.

> Well, they took care of me and put me on a stretcher, they checked the whole time that I was all right so it sure was more care than transport. (Ahl *et al*,[28] participant quotation)

The perceived absence of compassion when calling for an ambulance was also noteworthy, impacting on the overall experience.

> He didn't care. It was just when I was speaking to him he wasn't sort of listening. (Togher *et al*,[32] participant quotation)

Such codes occurred commonly with examples of clear mismatch between the ambulance clinician's assessment and the patient's assessment of medical need. In such circumstances, the emotional function of the ambulance service was not fulfilled, even if the medical function was.

### The process domain
Central to this analysis is the third analytical theme of the 'processes' involved in balancing the practical domain with the emotional domain. This analytical theme relates the emotional aspects of seeking urgent help with the practicalities of obtaining and receiving ambulance treatment by describing processes central to help seeking and decision making in this context. These 'process' descriptive themes were grouped into three: being influenced in the decision making by others, being empowered to take responsibility for one's health and taking or handing over control of the situation. This analysis demonstrated the frequent juxtaposition—even tension—of some of the themes related to practicalities with themes related to emotional responses.

A strong descriptive theme running through the analysis of the process domain was the idea of patients achieving some sense of control over a perceived urgent or distressing situation by phoning an ambulance:

> When patients phone for an ambulance, they hope to gain control of the experienced emergency situation based on trust and the expectation that they will be assisted… which implies an ethical demand. (Rantala *et al*,[30] author interpretation)

> It feels good once you have made the decision and phoned for an ambulance. They know that something is wrong… now I don't have to take further responsibility. (Ahl *et al*,[28] participant quotation)

The approach of decisively 'taking control' of the situation appears to be used by some patients to bring about a definitive resolution to an otherwise intolerable situation by accessing ambulance treatment. Contrastingly, some patients appear much more comfortable with handing over control of a situation they do not fully understand to the ambulance service, thereby exonerating themselves of responsibility for managing it. This concept of 'responsibility' appears quite complex. Linked with the emotional theme of reassurance is a need to also be 'reassured' that the decision to call an ambulance was correct—to seek legitimisation for using a finite (even scarce) resource for one's own needs. When callers 'hand over' this practical element of the decision-making by calling the emergency number, they appear to feel emotionally more comfortable with the outcome of an ambulance attending, as it is not directly they who made the decision to send one. Rather it is the system that has decided that an ambulance is necessary and is responsible for the outcome—there appears a degree of conceptual disconnect between calling for an ambulance and receiving one.

It is also clear how powerful the influence of others (particularly family) can also be on this particular aspect of effectively 'handing over' the decision-making process:

> It was my wife who was very concerned and wanted to make the call. (Rantala *et al*,[30] participant quotation)

A particularly important aspect of this descriptive theme is that the 'influencer' does not even necessarily have to be present at the time of the need for treatment for their views to form part of the decision-making.

> I could just hear my daughter saying 'oh mum, why didn't you get the ambulance out?' even though I didn't think I needed it. (Booker *et al*,[5] participant quotation)

Ambulance crews also have an ability to exert influence on the patients' decision-making, particularly around whether hospitalisation is necessary or not. Particularly noteworthy in the analysis is that the process of exerting this influence—while often conveyed in very practical terms (such as the likely lengthy waiting around in hospital for a bed to be available)—has the ability to produce a strong emotional response. This emotional response can often be one of empowerment and taking responsibility, through mutual respect, partnership and shared decision-making

> (Paramedics) also talked a great deal about the possibility of persuading people to stay at home when hospital treatment was not necessary. (Porter *et al,*[31] author interpretation)

> So I was respected as a full partner in the conversation. It was a very nice experience. (Rantala *et al,*[30] participant quotation)

But the process of influencing can also result in negative emotional responses such as guilt, which may even have direct consequences on the patient's choice of care in the future. Commonly the codes associated with negative influences occurred in encounters that were being labelled as 'inappropriate'.

> Afterwards I felt very guilty about what happened and why I had done it. I just hope I don't have to call them again. (Rantala *et al,*[30] participant quotation)

> One feels so worthless… they just don't believe me! (Rantala *et al,*[30] participant quotation)

## Discussion

This qualitative synthesis suggests that the decision-making processes involved in calling for and receiving ambulance treatment for a 'primary care sensitive' problem are shaped by a combination of emotional and practical factors. Often, the ambulance service contact functions to fulfil both emotional and practical needs in the caller. Regardless of the actual severity of the precipitating clinical condition, both of these groups of factors play an important role in choosing (and valuing) ambulance-delivered care. This synthesis proposes that three key processes are involved in combining emotional and practical factors: handing over of control and responsibility, the influencing by important others and empowerment to take responsibility for one's own health. Informal carers, relatives and health professionals can all be 'influencers' and do not even necessarily have to be physically present for their 'influence' to be considered by a patient.

There is currently discussion in the research and policy literature about how to define and reduce so-termed 'unnecessary' or 'inappropriate' ambulance use. This synthesis supports previous findings[5] that suggest the concept of 'inappropriate' ambulance use is far more nuanced than a definition based purely on the clinical condition that a patient receives treatment for. This analysis suggests there is a strong association

with negative emotional responses when callers believe they are being labelled as 'inappropriate'. The implications of these responses are complex, as there is a clear need to challenge inefficient (or irresponsible) use of the finite emergency health resources that are under increasing pressure and to promote personal health resilience strategies. However, this must be balanced with an understanding of what prompted the contact, so that additional barriers are not constructed that may adversely impact future help seeking for serious emergency situations. The themes from this review suggest that urgent care triage systems need to be sensitive to both the perceived emotional and practical drivers of a call for help but importantly need to recognise how this may be shaped by carers and relatives. While the ambulance service contact is sometimes therapeutic in itself, this synthesis highlights the ongoing conflict that ambulance services face with directing finite resources to resolving an individual's (potentially emotionally driven) need, verses the wider emergency care needs of the population. This synthesis suggests that the components of the treatment contact that patients value are those that support their own ability to 'process' both the emotional and practical challenges they are presented with by their health condition and enable them to find a comfortable level of control of the situation and possible outcomes. It is possible that these valued components might be deliverable in ways other than despatching emergency ambulances to physically attend.

## Conclusions

Supporting patients and their relatives to access the 'right advice in the right place, first time' has underpinned UK national urgent care policy for nearly a decade,[21] but is hampered by an incomplete understanding of how patients conceptualise urgency. This synthesis builds on recent work in the emergency department setting that suggested patients define situations worthy of emergency health resources according to socioemotional factors rather than purely the symptoms or physiology underlying their illness.[22] This synthesis also suggests that the 'organisation incentives' and 'secondary benefits' of the health system referred to in the classic sociological models of illness and help seeking[8] may also exist when choosing ambulance care via the processes of legitimisation and, perhaps somewhat paradoxically, empowerment. Previous work has acknowledged how practical issues such as rurality and remoteness (eg, Famer *et al* and Turnbill *et al*[23 24]), perceptions of out of hours primary care,(eg, Egbunike *et al* and Foster *et al*[25 26]) and views of telephone advice services[27] shape decision-making. This synthesis highlights how importantly patients regard the views of those around them and how emotionally conflicted some patients can be about their care choices.

## Practice implications

This qualitative systematic review is the result of a comprehensive, systematic literature search in accordance

with a prospectively published protocol. Although the resulting synthesis is based on a relatively small number of studies meeting the inclusion criteria, this reflects a relatively under-researched area of interest. The majority of included studies are from UK-based ambulance services, and all were undertaken in a European system with a primary healthcare model. As such, there may be some limitations in applicability to contrasting ambulance systems. Additionally, all reported studies were based on adult populations, limiting the conclusions that can be drawn for paediatric populations. Some of the included studies, while including relevant data about ambulance use for 'primary care sensitive' conditions, were not designed specifically to explore these groups. Despite these limitations, this study is, to the authors' knowledge, the first synthesis of qualitative evidence combining patients', carers' and health professionals' perspectives on the use of ambulances for lower acuity conditions. Future work needs to explore how urgent care systems can be sensitive and responsive to the emotional component of the help-seeking process while still providing the practical assistance that patients and carers require. This necessarily will involve a balanced approach that empowers urgent healthcare providers to tailor the format of the response accordingly. Additionally, with increasing pressures on ambulance services, there is a need for specific research exploring how to empower patients and carers towards resilient, resource-responsible help-seeking strategies in these situations.

**Contributors** MJB is the main author involved in all stages of the design, analysis and write-up. ARGS and SP contributed to the methodological approach, data extraction, verification and analysis through regular research meetings. ARGS and SP contributed significant input to the written paper. All authors have approved the final version of the paper ahead of publication.

**Funding** MJB is funded by a National Institute for Health Research Doctoral Research Fellowship. This paper presents independent research funded by the National Institute for Health Research (NIHR). The views expressed are those of the author(s) and not necessarily those of the NIHR, the NHS or the Department of Health.

**Competing interests** None declared.

**Provenance and peer review** Not commissioned; externally peer reviewed.

**Data sharing statement** No additional data are available.

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
