## [Reviewer comments · BMJ Open]

ARTICLE DETAILS

TITLE (PROVISIONAL)	Seeking ambulance treatment for 'primary care' problems: a qualitative systematic review of patient, carer and professional perspectives.
AUTHORS	Booker, Matthew; Purdy, Sarah; Shaw, Ali

VERSION 1 - REVIEW

REVIEWER	Alicia O'Cathain Uni of Sheffield UK Prof Sarah Purdy has chaired a Project Advisory Group for one of my projects. Dr Booker is a member of a Project Advisory Group for one of my projects
REVIEW RETURNED	23-Mar-2017

GENERAL COMMENTS	I think this is a useful and well conducted study in an under-researched area and makes a very helpful contribution to the evidence base. I appreciate the attention to methodology and the transparency of the analysis processes. It is a thoughtful analysis. Some issues need attention to improve the paper: I would like to see the abstract rewritten so that it better communicates the findings of the study. I think it draws heavily on the issue of judgement about inappropriateness; this was not a strongly displayed theme in the main body of the findings. It also relies on the use of words in inverted commas which are failing to communicate clearly the points being made. The discussion of inappropriate use of services in the later part of the paper could be revisited and a clearer argument made. It could be expanded in relation to population need versus individual need for health care provision in a context of limited resources. If someone calls an ambulance because they don't want to inconvenience their son and then get upset because the ambulance service has labelled them inappropriate then maybe this is OK. If it then affects their subsequent behaviour so they contact their son next time would we see this as a problem? Similarly around the emotional aspect of help-seeking, I think it is important to understand the powerful role of this, and the authors show this very well, but is the best solution to respond with an ambulance? that is, I would like to see further exploration of the implications of the findings.
--

	Some of the papers do not focus on the use of ambulance services for primary care sensitive conditions. There is no doubt that some of the sample within these papers fell into this group but the aim of the studies was not to explore this group and this is worth being clear about.
--	--

REVIEWER	Nigel Rees Pre Hospital Emergency Research Unit (PERU) Welsh Ambulance Services NHS Trust Wales Adl am the R&D lead for the Welsh Ambulance Services NHS Trust, I have fascilitated access to conduct research in our organisation
REVIEW RETURNED	29-Mar-2017

GENERAL COMMENTS	This is an important area of research to Ambulance Service providers, commissioners, clinicians and patients. Details are presented of the philosophical approach to this work, which is very important in such reviews which do not follow the traditional accumulation of logic approach adopted to systematic reviews in literature from the positivist realm. The author also presents a detailed audit trail of coding, decision making and synthesis. This again is reassuring to the reader, as it adds to the trustworthiness and transparency of the work. The author recognises the long held view of 'inappropriate' users of services, and by adopting this critical realist approach to synthesising the literature has offered an alternate consideration. This paper recognises complexity in the decision making of patients who call for ambulance services for primary care problems. The paper also recognises how labeling such ambulance services use as "inappropriate fails to recognise the context of the request for assistance, and is unhelpful for developing practical solutions". This understanding has the potential to positively influence high profile (and costly) public awareness campaigns such as "Choose well" Thanks for the opportunity to review this rich and detailed piece of work
---

VERSION 1 – AUTHOR RESPONSE

Reviewer 1:

I think this is a useful and well conducted study in an under-researched area and makes a very helpful contribution to the evidence base.

I appreciate the attention to methodology and the transparency of the analysis processes.

It is a thoughtful analysis.

Response: Thank you for these supportive comments.

Some issues need attention to improve the paper:

I would like to see the abstract rewritten so that it better communicates the findings of the study. I think it draws heavily on the issue of judgement about inappropriateness; this was not a strongly

displayed theme in the main body of the findings. It also relies on the use of words in inverted commas which are failing to communicate clearly the points being made.

Response: Thank you for highlighting that the abstract did not summarise the paper as well as it could have. In light of this, and the Editorial Team comments, this has been re-written to hopefully more clearly capture the main messages.

The discussion of inappropriate use of services in the later part of the paper could be revisited and a clearer argument made. It could be expanded in relation to population need versus individual need for health care provision in a context of limited resources. If someone calls an ambulance because they don't want to inconvenience their son and then get upset because the ambulance service has labelled them inappropriate then maybe this is OK. If it then affects their subsequent behaviour so they contact their son next time would we see this as a problem? Similarly around the emotional aspect of help-seeking, I think it is important to understand the powerful role of this, and the authors show this very well, but is the best solution to respond with an ambulance? that is, I would like to see further exploration of the implications of the findings.

Response: Thank you for drawing out this important aspect of the discussion. This has now been rewritten with a view to highlighting the tensions between individual needs and population needs, particularly in light of the recent attention being paid to added pressures on ambulance services. A key element of this analysis, we feel, is the importance of acknowledging to and responding to the emotional component driving the help-seeking behaviour, and we have tried to highlight this by rephrasing some of the discussion as you kindly suggest.

Some of the papers do not focus on the use of ambulance services for primary care sensitive conditions. There is no doubt that some of the sample within these papers fell into this group but the aim of the studies was not to explore this group and this is worth being clear about.

Response: Thank you for drawing our attention to the need to make this point very clear; we have added a line in the implications for practice section to explicitly state this.

Reviewer 2:

This is an important area of research to Ambulance Service providers, commissioners, clinicians and patients. Details are presented of the philosophical approach to this work, which is very important in such reviews which do not follow the traditional accumulation of logic approach adopted to systematic reviews in literature from the positivist realm. The author also presents a detailed audit trail of coding, decision making and synthesis. This again is reassuring to the reader, as it adds to the trustworthiness and transparency of the work.

The author recognises the long held view of 'inappropriate' users of services, and by adopting this critical realist approach to synthesising the literature has offered an alternate consideration. This paper recognises complexity in the decision making of patients who call for ambulance services for primary care problems. The paper also recognises how labeling such ambulance services use as "inappropriate fails to recognise the context of the request for assistance, and is unhelpful for developing practical solutions". This understanding has the potential to positively influence high profile (and costly) public awareness campaigns such as "Choose well"

Response: Thank you for these supportive comments regarding the methodology and conclusions.

VERSION 2 – REVIEW

REVIEWER	Alicia O'Cathain ScHARR, UK Dr Booker is on a project advisory group for one of my projects.
REVIEW RETURNED	16-Jun-2017

GENERAL COMMENTS	The authors have addressed the comments made at earlier review.
---